# Cost-effectiveness analysis of adding transarterial chemoembolisation to lenvatinib as first-line treatment for advanced hepatocellular carcinoma in China

Wei Li  , Li Wan

Department of Pharmacy, Maternal and Child Health Hospital of Hubei Province, Tongji Medical College, Huazhong University of Science and Technology, Wuhan, Hubei, China

**Correspondence to**
Dr Wei Li;
liwei1988hust@126.com

## ABSTRACT

**Objective** The objective of this study was to evaluate the comparative cost-effectiveness of lenvatinib (LEN) plus transarterial chemoembolisation (TACE) (LEN-TACE) and LEN alone to treat advanced hepatocellular carcinoma (HCC) from the perspective of the Chinese healthcare system.

**Design** A three-state partitioned survival model using clinical survival data from a phase III LAUNCH trial, a 5-year time horizon for costs and quality-adjusted life years (QALYs) was constructed to analyse the cost-effectiveness of LEN-TACE. Clinical inputs were extracted from the LAUNCH trial, with outcomes extrapolated using standard and flexible parametric survival models. Costs and utilities derived from published literature were discounted at an annual rate of 5%. Sensitivity analyses and scenario analyses were conducted to test the robustness of the model.

**Setting** The Chinese healthcare system perspective.

**Participants** A hypothetical Chinese cohort of patients with advanced HCC.

**Interventions** TACE plus LEN versus LEN.

**Primary outcome measure** Costs, QALYs, incremental cost-effectiveness ratio (ICER).

**Results** Base-case analysis revealed that LEN-TACE would be cost-effective in China at the willingness-to-pay (WTP) threshold of $37 663 per QALYs, with improved effectiveness of 0.382 QALYs and additional cost of $12 151 (ICER: $31 808 per QALY). The probabilistic sensitivity analysis suggested that LEN-TACE had a 93.5% probability of cost-effectiveness at WTP threshold of three times gross domestic product per capital ($37 663). One-way deterministic sensitivity analysis indicated that the duration of LEN treatment in both two arms, utility of progression-free survival and the cost of TACE had a greater impact on the stability of ICER values. Scenario analyses results were in line with base-case analysis.

**Conclusions** LEN-TACE might be a cost-effective strategy compared with LEN for the first-line treatment of patients with advanced HCC in China.

## STRENGTHS AND LIMITATIONS OF THIS STUDY

⇒ The partitioned survival model acquires the proportion of patients in different health states directly from the survival curves, without too many assumptions and is commonly applied in economic evaluations of advanced or metastatic cancer.

⇒ Both standard and flexible parametric survival models are used to predict the long-term survival outcomes in this cost-effectiveness analysis.

⇒ External data accessed through reconstructing the survival curves of previously clinical trials on hepatocellular carcinoma is used to verify our model choice.

⇒ In this model, the proportions of patients receiving subsequent anticancer treatment originated from clinical trial data, which might not true represent the prevalence of subsequent anticancer strategy used in real-world practice.

⇒ We do not conduct the subgroup analysis due to a short of subgroup survival data.

## INTRODUCTION

Hepatocellular carcinoma (HCC) is the sixth most commonly diagnosed cancer and ranks third in cancer-related mortality worldwide.[1] However, this situation is even worse in China, where it is the fifth most commonly diagnosed cancer and the secondary leading cause of cancer-related death.[2] It is estimated that more than 70% of HCC patients have advanced cancer at the time of diagnosis, which limits the feasibility of radical surgical treatment in advanced cases.[3 4] Therefore, targeted systemic therapies have become a ray of hope for advanced HCC patients. Lenvatinib (LEN), an oral multiple receptor tyrosine kinase inhibitor (TKI), non-inferior to sorafenib, is currently recommended as first-line treatments for advanced HCC.[5 6] Although the median overall survival (OS) for Chinese subgroup population was longer than that of the intention-to-treat (ITT) population (15.0 vs 13.6 months), the efficacy of LEN is still unsatisfactory.[7] To further

prolong the OS, much endeavours have been made, such as the combination of LEN with other treatments.

Transarterial chemoembolisation (TACE) can effectively reduce tumour burden by targeting intrahepatic tumours, thus may enhance the outcomes in patients with advanced HCC.[8] While, LEN is more likely to be effective in patients with lower tumour burden.[9] In terms of mechanism of action, the combination of LEN and TACE (LEN-TACE) has a synergistic effect. Recently, a series of observational studies indicate that the combination of LEN and TACE brings favourable survival outcomes.[10–12] However, these observation studies only provide limited supporting evidence and need to be further verified by large-scale clinical trial.

Nowadays, a Chinese-patient-based phase III trial, LAUNCH (NCT03905967), demonstrates that LEN-TACE prolongs median OS by a significant 6.3 months, compared with LEN monotherapy in the ITT population.[13] Additionally, LEN-TACE shows a statistically significant improvement in progression-free survival (PFS) time, with a median improvement in PFS of 4.2 months.[13] Although the remarkable results of this trial raises hope for patients extend survival, higher medical costs generated by combination therapy may impose a heavy socioeconomic burden on patients and the healthcare system, especially in resource-limiting settings like China. At present, economic evaluations of LEN in the area of HCC mainly focus on the comparision of intervention regimes, such as the LEN versus LEN similar drugs (sorafenib, donafenib), LEN versus sintilimab plus bevacizumab.[14–16] There is no cost-effectiveness evaluation of LEN-TACE strategy in patients with advanced HCC from the perspective of Chinese healthcare system. We thus compare the cost-effectiveness of the two strategies to treat advanced HCC by using model data from LAUNCH. These findings provide evidence for use by HCC patients and the physicians treating them, as well as health policymakers.

## METHODS
### Model structure and outcomes
A partitioned survival model, including three mutually exclusive disease-related health states: PFS, progressed disease (PD) and death, was constructed in TreeAge Pro 2020 (TreeAge Software, Williamstown, MA) to simulate the cohort of patients in LAUNCH trial.[13] The model cycle length was a treatment cycle (1 month). The time horizon was set as 5 years, with 99% of people dying. This analysis was performed from the perspective of Chinese healthcare system. Our target population was patient with primary advanced HCC receiving first-line treatments in China. These patients underwent LEN alone or plus TACE (LEN-TACE) during the simulated time. The tree diagram and bubble diagram were illustrated in online supplemental figure S1. It was hypothesised that the patient cohort entered the model in the PFS state and then either occupied in the same state or moved to the other states according to transition probabilities during

each model cycle. The initial age of the simulated cohort population was set to 55 years old, which was consistent with the average age of the LAUNCH trial.[13]

The primary model outcomes were the corresponding total costs of two therapeutic regimens, quality-adjusted life years (QALYs), and the incremental cost-effectiveness ratios (ICERs). Both costs and utility values were discounted at 5% annually for base-case analysis, according to the guideline for health economic evaluations in China.[17] All costs were converted into 2021 US dollars (US1\$=¥6.45). Threefold of the per capita gross domestic product (GDP) of China in 2021 (US\$37 663/QALYs) was used as the willingness-to-pay (WTP) threshold to evaluate the cost-effectiveness of the two competing strategies.[17]

### Clinical data and model probabilities
The clinical efficacy and safety data of LAUNCH trial, which was conducted at 12 hospitals in China, were used to explore the cost-effectiveness of LEN with or without TACE for advanced HCC.[13] The PFS and OS data were extracted from the published K-M curves by using GetData Graph Digitizer software (V.2.26) and then was used to reconstruct the individual patient data (IPD) according to Guyot's method.[18] To extrapolate survival outcomes outside the observation period, different parameter distributions (exponential, Weibull, Gompertz, log-logistic, log-normal, gamma, gen-gamma and Royston/Parmar spline model) were employed to fit the reconstructed IPD. The choice of survival model to use was based on statistical goodness of fit (using the Akaike's information criterion (AIC) and the Bayesian information criterion (BIC)), visual inspection and clinical plausibility of the extrapolations. Finally, the log-logistics model had the best AIC and BIC was used for both arms for PFS and OS in base-case analysis (see online supplemental table S1). We also provided the exploration and fitting of PFS and OS curves for visual inspection (see online supplemental figure S2). The choice of log-logistic model as survival distribution for extrapolation in base-case analysis was consistency with previous findings of extrapolation in advanced HCC.[19] The log-logistic model used for OS in LEN-TACE arm estimated a 3-year survival rate of 9.2% compared with that of 9.5%, reported by a retrospective study (median follow-up time, 27 months) in China.[20] Although the log-logistic model used for OS in LEN arm predicted a 3-year survival rate of 3.6%, which was not consistent with that of 13% reported in REFLECT trial, we thought that this model exhibited a reasonable fit to the observed data.[21] The reasons were as follows: (a) there were some differences in baseline patient characteristics between LAUNCH and REFLECT trial, especially in α-fetoprotein and Barcelona Clinic Liver Cancer stage; (b) the higher α-fetoprotein level and more advanced Barcelona Clinic Liver Cancer stage in LAUNCH trial might be associated with lower OS rate. The impact of selecting alternative survival model was investigated in scenario analyses.

**Table 1** Model parameters

| Item | Base case | Range | Distribution | Source |
|---|---|---|---|---|
| **Costs (US$)** | | | | |
| Lenvatinib per 120 mg (PATHEONINC) | 499.71 | 399.77–499.71 | Gamma | 25 |
| Regorafenib per 1120 mg (Bayer AG) | 744.85 | 372.43–744.85 | Gamma | 25 |
| Tislelizumab per 100 mg (BeiGene) | 223.63 | 178.91–223.63 | Gamma | 25 |
| TACE per session | 1929 | 1543–2315 | Gamma | 24 |
| Hepatectomy | 9022 | 7218–10 827 | Gamma | 24 |
| **Management of adverse events** | | | | |
| Elevated ALT/AST | 56.54 | 45.23–67.84 | Gamma | 23 |
| Hypertension | 35.46 | 28.36–42.55 | Gamma | 23 |
| Hyperbilirubinaemia | 113.53 | 90.82–136.24 | Gamma | 15 |
| Diarrhoea | 5.66 | 4.53–6.79 | Gamma | 22 |
| Decreased weight | 102.73 | 98.67–120.63 | Gamma | 15 |
| BSC cost per cycle | 265.08 | 212.06–318.10 | Gamma | 25 |
| Follow-up and monitoring per month in PFS | 114 | 86–143 | Gamma | 25 |
| Follow-up and monitoring per month in PD | 210 | 157–262 | Gamma | 25 |
| Terminal care per patient | 1839 | 1519–2279 | Gamma | 25 |
| **Utility values** | | | | |
| PFS | 0.76 | 0.61–0.91 | Beta | 28 |
| PD | 0.68 | 0.54–0.82 | Beta | 28 |
| **Disutility of adverse events** | | | | |
| Elevated ALT/AST | 0 | NA | NA | 15 |
| Hypertension | 0.012 | 0.010–0.014 | Beta | 15 |
| Hyperbilirubinaemia | 0 | NA | NA | 15 |
| Diarrhoea | 0.047 | 0.016–0.077 | Beta | 22 |
| Decreased weight | 0.053 | 0.042–0.064 | Beta | 15 |
| **LEN-TACE: incidence of SAEs** | | | | |
| Elevated ALT/AST | 0.40 | 0.32–0.48 | Beta | 13 |
| Hypertension | 0.20 | 0.16–0.24 | Beta | 13 |
| Hyperbilirubinaemia | 0.09 | 0.072–0.108 | Beta | 13 |
| Diarrhoea | 0.05 | 0.04–0.06 | Beta | 13 |
| Decreased weight | 0.076 | 0.06–0.09 | Beta | 13 |
| **LEN: incidence of SAEs** | | | | |
| Elevated ALT/AST | 0.03 | 0.024–0.036 | Beta | 13 |
| Hypertension | 0.20 | 0.16–0.24 | Beta | 13 |
| Hyperbilirubinaemia | 0.03 | 0.024–0.036 | Beta | 13 |
| Diarrhoea | 0.04 | 0.032–0.048 | Beta | 13 |
| Decreased weight | 0.07 | 0.056–0.084 | Beta | 13 |
| Discount rate (%) | 5 | 0–8 | Fixed in PSA | |
| **Proportion of receiving subsequent treatments in LEN-TACE** | | | | |
| Hepatectomy | 0.16 | 0.128–0.192 | Beta | 13 |
| Regorafenib | 0.22 | 0.176–0.264 | Beta | 13 |
| Tislelizumab | 0.43 | 0.344–0.516 | Beta | 13 |
| BSC | 0.19 | 0.152–0.228 | Beta | 13 |
| **Proportion of receiving subsequent treatments in LEN** | | | | |
| TACE | 0.07 | 0.056–0.084 | Beta | 13 |
| Regorafenib | 0.25 | 0.20–0.30 | Beta | 13 |

Continued

**Table 1** Continued

| Item | Base case | Range | Distribution | Source |
|---|---|---|---|---|
| Tislelizumab | 0.51 | 0.408–0.612 | Beta | 13 |
| BSC | 0.17 | 0.136–0.204 | Beta | 13 |
| LEN treatment duration (months) | 15.0 | 8.0–15.0 | Normal | 13 |

ALT, alanine aminotransferase; AST, aspartate aminotransferase; BSC, best supportive care; LEN, lenvatinib; PD, progressed disease; PFS, progression-free survival; PSA, probabilistic sensitivity analysis; SAEs, serious adverse events; TACE, transarterial chemoembolisation.

## Cost and utility

Only direct medical costs, including the first-line and subsequent treatment cost, monitoring cost, hepatectomy cost, TACE cost, management of serious adverse events (SAEs, grade 3–4), follow-up cost and cost of terminal care in end of life, were calculated from the perspective of the Chinese healthcare system. It should be noted that follow-up costs included CT examination, urinalysis, blood test and blood biochemical examination; monitoring costs included diagnosis fee, nursing fee, injection fee and bed fee, more details were presented in table 1. Based on the LAUNCH trial, the median number of TACE sessions per patient was 3 (range, 1–6 sessions), once every 8 weeks.[13] All costs were derived from previously published literature.[22–25] To calculate the costs in base-case analysis, we made the following assumptions in this model:

1. We assumed that the average weight of a patient was 60 kg.
2. We did not consider any cause of dose reduction or interruption of LEN in base-case analysis. Patients were assumed to receive full dose LEN (body weight ≥60 kg, 12 mg daily; body weight <60 kg, 8 mg daily) until disease progression or unacceptable toxicity.
3. In line with the LAUNCH trial protocol, patients after progression or discontinuation would receive subsequent treatments, such as curative surgical resection, TKI therapy, programmed cell death protein-1 (PD-1) inhibitor. The selection of specific TKI and PD-1 inhibitor as subsequent treatment was based on the guideline of China.[26] For simplification, we assumed that patients in LEN-TACE arm would receive one of the following subsequent treatments: hepatectomy, regorafenib (160 mg/day, 3 weeks of medications, then discontinuing for 1 week), tislelizumab (200 mg/day, q3w), best supportive care (BSC); patients in LEN arm would receive one of the following subsequent treatments: regorafenib (160 mg/day, 3 weeks of medications, then discontinuing for 1 week), tislelizumab (200 mg/day, q3w), TACE, BSC. The proportion of patients receiving subsequent therapy was derived from the LAUNCH trial.
4. To better reflect the cost of real-world clinical practice, the duration of treatment in PFS state was considered. The cost for LEN was charged based on the upper limit of the treatment duration, as per the LAUNCH trial. In addition, a 2-year maximum treatment duration of tislelizumab was taken into consideration based on previous study.[27]
5. Only grade 3 or 4 SAEs with an incidence of >5% were considered, including elevated alanine aminotransferase (ALT)/aspartate aminotransferase (AST), hypertension, decreased weight, diarrhoea and hyperbilirubinaemia. The 3–4 SAEs-related costs were calculated once in the first cycle by multiplying the incidence of the SAEs by the costs of managing the SAEs per event.

**Table 2** Base-case and scenario analyses results

| Treatment | Cost ($) | ΔCost | QALYs | ΔQALYs | ICER |
|---|---|---|---|---|---|
| **Base-case** | | | | | |
| LEN-TACE | 31 394 | 12 151 | 1.166 | 0.382 | 31 808 |
| LEN | 19 243 | NA | 0.784 | NA | NA |
| **Scenario** | | | | | |
| Weibull model | | | | | |
| LEN-TACE | 30 629 | 11 736 | 1.072 | 0.328 | 35 780 |
| LEN | 18 893 | NA | 0.744 | NA | NA |
| Royston/Parmar spline model | | | | | |
| LEN-TACE | 31 468 | 10 733 | 1.203 | 0.320 | 33 540 |
| LEN | 20 735 | NA | 0.883 | NA | NA |

ICER, incremental cost-effectiveness ratio; LEN, lenvatinib; NA, not applicable; QALYs, quality-adjusted life years; TACE, transarterial chemoembolisation.

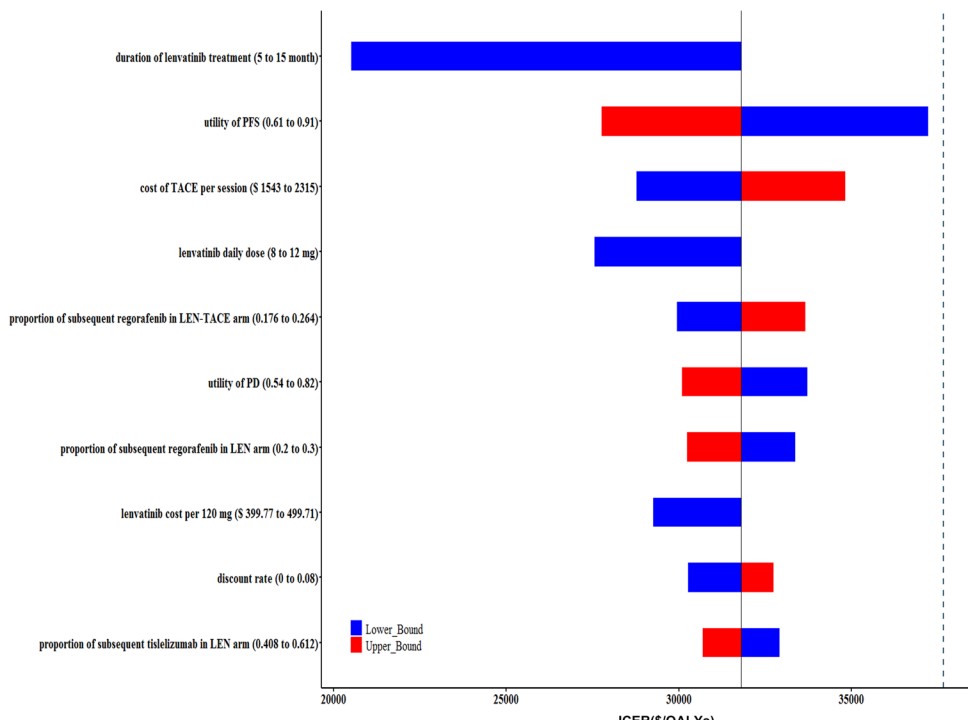

**Figure 1** Tornado diagram of one-way deterministic sensitivity analysis (DSA) of LEN-TACE versus LEN. The dashed line represents the threshold of willingness-to pay ($37 663/QALYs). ICER, incremental cost-effectiveness ratio; LEN, lenvatinib; PD, progressed disease; PFS, progression-free survival; QALYs, quality-adjusted life years; TACE, transarterial chemoembolisation.

6. Given that the subsequent treatment information was limited from the LAUNCH trial records. The grade 3–4 SAEs-related costs for subsequent treatment were ignored.

7. We assumed that all the patients received terminal care 3 months before they died in the base-case analysis.

Health state utility values were derived from previously published cost-effectiveness studies regarding Chinese patients with unresectable HCC, and the values were set at 0.76 for the PFS, 0.68 for PD state, respectively.[28] Considering that the occurrence of adverse events might have impact on patients' health-related quality of life. This

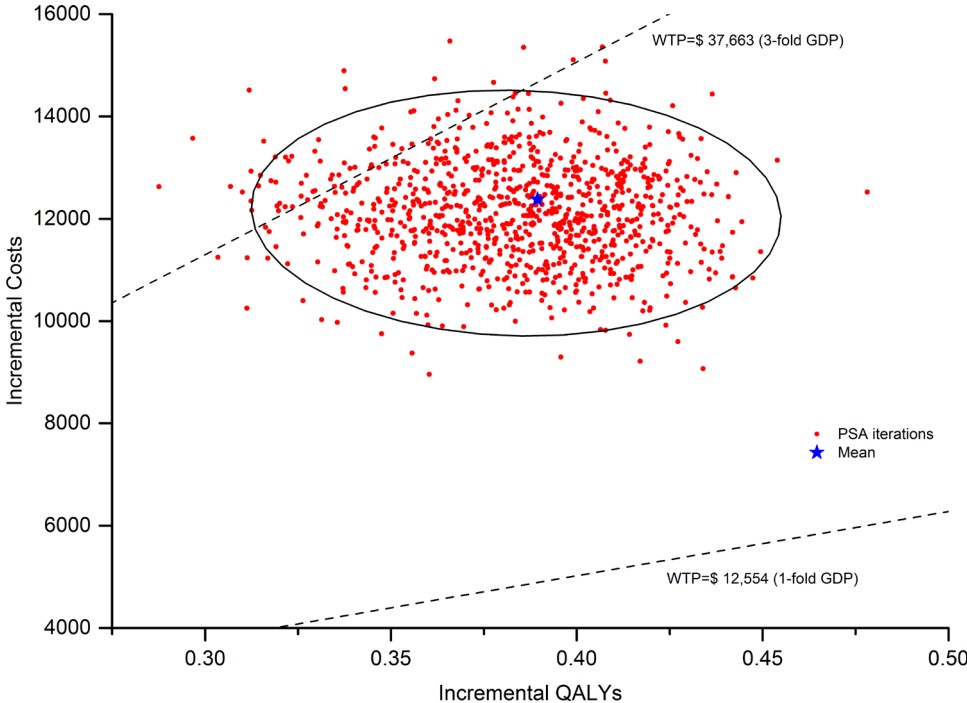

**Figure 2** Probabilistic sensitivity analysis (PSA) scatter plot (1000 iterations). An ellipse surrounds 95% of the estimates. GDP, gross domestic product; QALYs, quality-adjusted life years; WTP, willingness-to-pay.

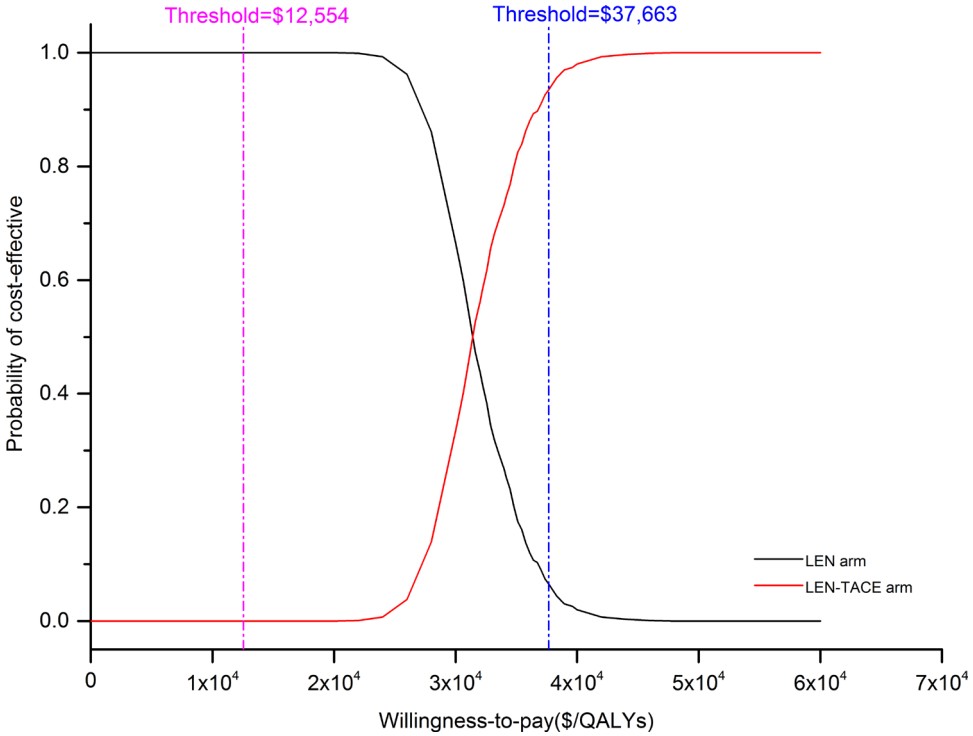

**Figure 3** Cost-effectiveness acceptability curve of LEN-TACE versus LEN. LEN, lenvatinib; QALYs, quality-adjusted life years; TACE, transarterial chemoembolisation.

present study also considered the disutility due to adverse events, including elevated ALT/AST, hypertension, decreased weight, diarrhoea and hyperbilirubinaemia, which were obtained from previous researches.[15 22] The details were listed in table 2.

### Sensitivity analysis and scenario analysis

One-way deterministic (DSA) and probabilistic sensitivity analysis (PSA) were performed to examine the robustness of model outcomes. To identify factors that had substantial impact on ICER, all variables varied over the plausible ranges, which were obtained from their corresponding CIs or ±20% of the base-case values (table 1). It should be noted that a body weight less than 60 kg associated with lower LEN dose (8 mg daily), resulting in the difference in treatment costs should not be ignored. Hence, we assumed an HCC patient weighted less than 60 kg (LEN, 8 mg daily) to investigate the impact on model outcomes. The results of DSA were graphed in the tornado diagram. PSA was performed to determine the effects of uncertainty in all model parameters simultaneously varied with prespecified distributions via 1000 Monte Carlo simulations, which were illustrated in scatter plot and cost-effectiveness acceptability curves (CEACs). Specifically, utility values, probabilities or proportions were assigned beta distributions, costs and treatment duration were apportioned to gamma distribution and normal distributions, respectively. The key parameters input our model were shown in table 1 and online supplemental table S1.

We also conducted two scenario analyses to explore the impact of different key model setting and assumptions on the economic outcomes. To eradicate the uncertainty

caused by the extrapolation of the survival curves, we used different parametric distribution survival models (ie, Weibull and Royston/Parmar spline model) to estimate the ICER. The Weibull distribution obtained lower survival benefit compared with that of other distributions in this study, so it was suitable to examine the model stability under extreme conditions. The parameter values of Weibull distribution and Royston/Parmar spline model for PFS and OS curves were exhibited in online supplemental table S1.

### Patient and public involvement

No patients or public were involved in the study.

## RESULTS
### Base-case analysis

In our base-case analysis, LEN-TACE cost $31 394 and yielded 1.166 QALYs, whereas LEN cost $19 243 and yielded 0.784 QALYs (see table 2). The ICER of LEN-TACE versus LEN was $31 808 per QALY, which was below the WTP threshold ($37 663/QALYs), demonstrating that the LEN-TACE was a cost-effective treatment option for advanced HCC.

### Sensitivity analysis and scenario analysis

DSA result indicated that the model was most sensitive to the duration of LEN treatment in both two arms, utility of PFS, and the cost of TACE. We only presented the top 10 sensitive factors in figure 1. All parameters fluctuated within the range in the DSA did not bring the ICER values surpass the WTP threshold ($37 663/QALYs). When the

duration of LEN treatment in both two arms increased from 5 to 12 months in this model, the ICER went up from \$20 508 to \$31 808 per QALYs correspondingly. For patient with a body weight <60 kg, namely the LEN daily dose reduced from 12 to 8 mg, the ICER dropped from \$31 808 to \$27 559 per QALYs. In other words, for patients with lower body weight, LEN-TACE had more favourable economic outcomes in comparison with LEN. As shown in figure 2, most of scatter points were between the line of onefold and threefold GDP. The CEACs also indicated that the probability of LEN-TACE being cost-effective was about 93.5% at the threshold of \$37 663/QALYs, which was consistent with those of base-case analysis outcomes (figure 3).

Assuming alternative Weibull and Royston/Parmar spline model would lead to the slight increase of ICER values (\$35 780, 33 540 vs \$31 808) compared with that of log-logistic model, respectively (table 2). Overall, the results of scenario analyses were congruous with the conclusions of the base-case analysis, confirming the robustness of our model outcomes. The CEACs of scenario analyses were given in online supplemental figures S3 and S4.

### Model validation

External data accessed through reconstructing the survival curves of previously clinical trials on HCC was used to verify our model choice.[20 21] Validation results for modelled PFS and OS data were presented in online supplemental table S2 and figure S2. The 6, 12, 24, 36-month survival rate estimated by best-fitting distribution and alternative models compared with the results that observed in LAUNCH trial and external data were presented in online supplemental table S2 and figure S2.

### DISCUSSION

To our knowledge, this is the first study to assess the economic outcomes of adding TACE to LEN as a first-line treatment for advanced HCC in China. The findings of this study suggested that the addition of TACE to LEN, as an alternative treatment option for Chinese patients with advanced HCC, might shed light on a potentially cost-effective practice-changing opportunity for clinicians, researchers and policymakers in the future.

In this study, we demonstrated that LEN-TACE might be a cost-saving treatment option for advanced HCC patients, with ICER versus LEN of \$31 808 per QALYs. PSA results suggested that the probability of LEN-TACE being cost-effective was 93.5% at the WTP threshold of \$37 663/QALYs, which proved the robustness of the base-case analysis results. When choosing alternative survival models, such as Weibull and Royston/Parmar spline model, the conclusion of LEN-TACE strategy being cost-effective was unchanged with a probability of 68.4% and 83.0%, respectively (online supplemental figures S3 and S4). The DSA results revealed that the ICER was most sensitive to the duration of LEN treatment in both two

arms, utility of PFS, and the cost of TACE. However, none of parameter variation could lead to an inversion of this economic evaluation results. It also should be noted that the utility value of PFS varying within a relatively wide range (0.61–0.91) would lead to a very close to the threshold of WTP. However, if we input a narrowed PFS utility value variation range (0.73–0.76) as reported by National Institute for Health and Care Excellence single technology appraisal in our model, the obtained ICER (\$31 804–32 759/QALYs) was significantly below the threshold line.[29] In summary, the outcomes of this economic evaluation model were reliable.

A number of existing limitations in this study should be noted. First, due to the lack of utility based on only local populations, the EQ-5D utilities of the PFS and PD states deriving from previously published cost-effectiveness analysis considering advanced HCC were used in this model, which may bring bias. We will update our model outcomes when these data are available in China. Second, the health outcomes beyond the follow-up period were estimated by parametric distributions, and the selection of the best fit distribution largely depend on the long follow-up external data. Inappropriate choice of parameter models would result in overestimation or underestimation of survival rate, which in turn, brought bias to the economic evaluation. Third, the proportions of patients receiving subsequent anticancer treatment originated from clinical trial data, which might not true represent the prevalence of subsequent anticancer strategy used in real-world practice. Fourth, we did not conduct the subgroup analysis due to a short of subgroup PFS and OS data. We will continue to investigate subgroups patients in the future when more data on subgroup populations become available.

### CONCLUSIONS

The findings of this economic evaluation suggest that the combination of LEN and TACE is likely to be a cost-effectiveness treatment option for Chinese patients with advanced HCC under the WTP threshold of \$37 663 in China. In the future, further long-term follow-up data and real-world data are needed to verify the robustness of model outcomes.

**Contributors** WL contributed to the study concepts and design, analysis and interpretation of the data, drafting and revising of the paper. LW contributed to the review of the paper and supervision. All authors approved of final manuscript. All authors agree to be accountable for all aspects of the work. WL, guarantor.

**Funding** This work was supported by the Maternal and Child Health Hospital of Hubei Province Research Project (No. 2021SFYM030).

**Competing interests** None declared.

**Patient and public involvement** Patients and/or the public were not involved in the design, or conduct, or reporting, or dissemination plans of this research.

**Patient consent for publication** Not applicable.

**Ethics approval** Not applicable.

**Provenance and peer review** Not commissioned; externally peer reviewed.

**Data availability statement** Data are available upon reasonable request.

**ORCID iD**
Wei Li http://orcid.org/0000-0001-9123-7908

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
