## [Reviewer comments · BMJ Open]

ARTICLE DETAILS

TITLE (PROVISIONAL)	Cost-effectiveness analysis of adding transarterial chemoembolization to lenvatinib as first-line treatment for advanced hepatocellular carcinoma in China
AUTHORS	Li, Wei; Wan, Li

VERSION 1 – REVIEW

REVIEWER	Chiang, Chi Leung University of Hong Kong, Department of Clinical Oncology
REVIEW RETURNED	22-May-2023

GENERAL COMMENTS	This is a nicely conducted cost-effectiveness analysis of TACE+Len vs. Len in patients with advanced HCC based on LAUNCH trial in Chinese societal perspective 1) Please justify the use of partitioned survival model instead of Markov model2) Please reorganise the discussion into a more concise and precise manner3) Please specify all the parameters tested in one-way sensitivity analyses and rationale in choosing these parameters Minor comments 1) Typo on table 1 Rego cost
--

REVIEWER	Arabloo, Jalal Iran University of Medical Sciences, Health management and Economics Research Center
REVIEW RETURNED	19-Jun-2023

GENERAL COMMENTS	The study presents a well-designed economic evaluation of the cost-effectiveness of LEN-TACE compared to LEN alone for the treatment of advanced HCC from the perspective of the Chinese healthcare system. The authors used a three-state partitioned survival model and conducted sensitivity analyses to test the robustness of the model. The results suggest that LEN-TACE might be a cost-effective strategy for the first-line treatment of patients with advanced HCC in China. I have some minor comments: It would also be useful to suggest subgroup analysis in future studies to determine the cost-effectiveness of LEN-TACE for specific patient populations. Are the cost ranges of Lenvatinib (120 mg) from PATHEONINC and Regorafenib (1120 mg) from Bayer AG accurate? Could you please verify them again? Authors mentioned "At present, economic evaluations of LEN in the area of
--

	HCC mainly focus on the comparison of intervention regimes, such as the LEN vs. LEN similar drugs (sorafenib, donafenib), LEN vs. sintilimab plus bevacizumab. There is no cost-effectiveness evaluation of LEN-TACE strategy in patients with advanced HCC from the perspective of Chinese healthcare system". Could you please provide an explanation of the current standard of treatment based on national and international guidelines? Additionally, could you clarify why the two treatments being compared in this study, LEN-TACE and LEN alone, have not been compared before in the literature?
--	---

VERSION 1 – AUTHOR RESPONSE

Reviewer: 1

Dr. Chi Leung Chiang, University of Hong Kong

Comments to the Author:

This is a nicely conducted cost-effectiveness analysis of TACE+Len vs. Len in patients with advanced HCC based on LAUNCH trial in Chinese societal perspective

1) Please justify the use of partitioned survival model instead of Markov model

Response: Thank you for your comment. As a fact, increasing number of pharmaco-economic literatures tend to use the partitioned survival model (PSM) for advanced/unresectable HCC (Su D et al, 2021, JAMA Netw Open; Saiyed M et al, 2020, Clin Drug Investig). In comparison with Markov model, the PSM do not need to calculate the transition probability and avoid the need for additional hypotheses, such as whether death is permitted from all health states (Hoyle M et al, 2011, Value Health). Therefore, we select PSM instead of Markov model in this study.

2) Please reorganise the discussion into a more concise and precise manner

Response: Thank you for your suggestion. We have reorganised the discussion into a more concise and precise manner in our revised version. We adjusted the order of contents in the discussion section and moved forward the parametric distribution model selection in discussion section to the method section. Hope you are satisfied with this reply.

3) Please specify all the parameters tested in one-way sensitivity analyses and rationale in choosing these parameters

Response: Thank you for your comment. We have specified the ranges and distributions of all the parameters tested in sensitivity analyses in our revised version. These parameters mainly have effect on the calculation of costs and QALYs, thus are involved in sensitivity analyses.

Minor comments

1) Typo on table 1 Rego cost

Response: Thank you for your comment. We are so sorry that we do not catch your meaning. The base case, range, and distribution of the regorafenib costs were derived from published study. After carefully checked the typo of cost in Table 1, we still do not find the problem. We just use the comma as the thousand separator in our revised version. If there is any discrepancy in my response, please inform me, and I will modify it in the next revision.

Reviewer: 2

Dr. Jalal Arabloo, Iran University of Medical Sciences

Comments to the Author:

The study presents a well-designed economic evaluation of the cost-effectiveness of LEN-TACE compared to LEN alone for the treatment of advanced HCC from the perspective of the Chinese healthcare system. The authors used a three-state partitioned survival model and conducted sensitivity analyses to test the robustness of the model. The results suggest that LEN-TACE might be a cost-effective strategy for the first-line treatment of patients with advanced HCC in China. I have some minor comments:

(1) It would also be useful to suggest subgroup analysis in future studies to determine the cost-effectiveness of LEN-TACE for specific patient populations.

Response: It is indeed a good suggestion. Although the LAUNCH trial displays the PFS and OS of subgroup analyses, the only available outcomes for subgroup analysis were HR. Inability to obtain other data, such as adverse events data, PFS and OS curves, may affect model establishment. We will continue to investigate subgroups patients in the future when more data on subgroup populations become available. Hope you are satisfied with this reply.

(2) Are the cost ranges of Lenvatinib (120 mg) from PATHEONINC and Regorafenib (1120 mg) from Bayer AG accurate? Could you please verify them again?

Response: Thank you for your comment. The cost ranges of lenvatinib and regorafenib are obtained from a published health technology assessment study that conducted in China. In sensitivity analysis, the upper boundaries for lenvatinib and regorafenib are the same as that of the base case value. This is due to the China's National Healthcare Security Administration (NHSA) negotiations on drugs, which leads dramatic drug price reduction. So, the upper boundaries for these drugs are the latest local public bid-winning price and the lower boundaries are 80% of the upper boundaries/base case value. By querying public database again, we confirm the cost ranges are accurate.

(3) Authors mentioned "At present, economic evaluations of LEN in the area of HCC mainly focus on the comparison of intervention regimes, such as the LEN vs. LEN similar drugs (sorafenib, donafenib), LEN vs. sintilimab plus bevacizumab. There is no cost-effectiveness evaluation of LEN-TACE strategy in patients with advanced HCC from the perspective of Chinese healthcare system". Could you please provide an explanation of the current standard of treatment based on national and international guidelines? Additionally, could you clarify why the two treatments being compared in this study, LEN-TACE and LEN alone, have not been compared before in the literature?

Response: Thank you for your comment. According to the NCCN and CSCO guidelines, the first line treatment options for advanced HCC mainly include multi-targeted kinase inhibitors (MTIs), immune checkpoint inhibitors (ICI), and the combination of MTIs and ICI. LEN plus TACE has not been recommended by Chinese guidelines as a first-line treatment for advanced HCC due to the lack of high-level evidence-based evidence. Fortunately, the Chinese population-based LAUNCH trial first reports that LEN-TACE prolong PFS and OS in comparison with LEN alone, which will provide more options for clinical treatment decision. In addition, both retrospective controlled study (Fu Z et al, 2021, Hepatol Int) and prospective cohort study (Yang B et al, 2021, Front Oncol) have demonstrated the safety and efficacy of LEN-TACE. As more and more evidences emerge, it is only a matter of time before this combination strategy been recommended by the Chinese guidelines. By searching literature, we find that no economic evaluation on LEN-TACE base on LAUNCH trial. This study is a trial-based cost effectiveness analysis, so we select LEN alone as comparator. Hope you are satisfied with this reply.